# Hormone Receptor Expression Variations in Normal Breast Tissue: Preliminary Results of a Prospective Observational Study

**DOI:** 10.3390/jpm11050387

**Published:** 2021-05-08

**Authors:** Giacomo Santandrea, Chiara Bellarosa, Dino Gibertoni, Maria C. Cucchi, Alejandro M. Sanchez, Gianluca Franceschini, Riccardo Masetti, Maria P. Foschini

**Affiliations:** 1Clinical and Experimental Medicine PhD Program, University of Modena and Reggio Emilia, 41121 Modena, Italy; giacomo.santandrea@ausl.re.it; 2Pathology Unit, Azienda USL-IRCCS di Reggio Emilia, 42122 Reggio Emilia, Italy; 3Unit of Anatomic Pathology at Bellaria Hospital, Department of Biomedical and Neuromotor Sciences, University of Bologna, 40126 Bologna, Italy; chiara.bellarosa@studio.unibo.it; 4Unit of Hygiene and Biostatistics, Department of Biomedical and Neuromotor Sciences, University of Bologna, 40126 Bologna, Italy; dino.gibertoni2@unibo.it; 5Breast Surgery Unit, Bellaria Hospital, AUSL Bologna, 40126 Bologna, Italy; mariacristina.cucchi@ausl.bo.it; 6Multidisciplinary Breast Center–Dipartimento Scienze della Salute della donna e del Bambino e di Sanità Pubblica, Fondazione Policlinico Universitario A. Gemelli IRCCS, 00168 Rome, Italy; gianluca.franceschini@policlinicogemelli.it (G.F.); riccardo.masetti@policlinicogemelli.it (R.M.)

**Keywords:** breast cancer, normal breast, breast pathology, hormone receptor, hormone expression

## Abstract

Normal breast tissue undergoes great variations during a woman’s life as a consequence of the different hormonal stimulation. The purpose of the present study was to examine the hormonal receptor expression variations according to age, menstrual cycle, menopausal state and body mass index. To this purpose, 49 tissue samples of normal breast tissue, obtained during surgery performed for benign and malignant conditions, were immunostained with Estrogen (ER), Progesterone (PR) and Androgen receptors (AR). In addition, Ki67 and Gross Cystic Disease Fluid Protein were studied. The data obtained revealed a great variability of hormone receptor expression. ER and AR generally increased in older and post-menopausal women, while young women presented a higher proliferative rate, evaluated with Ki67. PR increase was observed in women with BMI higher than 25. The different hormonal receptor expression could favor the development of breast cancer.

## 1. Introduction

Physiological variations in the expression of Estrogen receptor alpha (ER) and Progesterone receptor (PR) play an important role in breast in breast remodeling during physiological changes: from embryological development to puberty [1] as well as during menstrual cycle, pregnancy and even after menopause [2]. 

As it happens for physiological variations of breast glandular tissue, the expression of hormonal receptors is thought to be an underlying mechanism involved in breast cancer onset, with determinant variations induced by well-known risk factors, such as age [3,4] exogenous hormone use [5] or Body Mass Index (BMI) [6,7]. Indeed, breast cancer is now classified according to a combination of hormone receptor expression, Ki67 labelling index and HER2 [8,9]. 

Tot [10,11] proposed the theory of the “sick lobe”, according to which breast cancer arises in a genetically predisposed breast epithelium. Tot based his theory on cytokeratin expression; nevertheless, hormone receptor expression variations occurring during life could predispose the breast epithelium to malignant transformation. 

Furthermore, breast cancer presents differences in young pre-menopausal women and in older post-menopausal women [12], with ER negative cases being more frequently in the young.

Among hormonal receptors that are normally expressed in breast tissue, prior studies confirmed that the expression of ER and PR may be associated with subsequent breast cancer risk [5,13,14,15,16,17,18]. However, there is still scarce evidence regarding a larger panel of breast tissue receptors, including Androgen Receptor (AR), Gross Cystic Disease Fluid Protein 15 (GCDFP-15) and Ki67. 

The aim of this study is to investigate the expression of ER, PR, AR, GCDFP-15 and Ki67 in breast normal tissue according to age, BMI, menstrual cycle and the onset of a breast neoplasm (benign vs. malignant). 

## 2. Materials and Methods

### 2.1. Patients Selection

All patients who underwent surgery for benign or malignant breast lesions between June 2015 and January 2016 at the Breast Surgery Unit of Bellaria Hospital (Bologna, Italy), were asked to participate to the present study. Seventy-nine patients accepted.

Among them, 2 pre-menopausal patients who experienced post-chemotherapy menstrual cycle arrest and 28 patients who had only a little amount of normal glandular tissue, insufficient for the analyses, were excluded.

The 49 remaining patients constituted the study population and were grouped as follows:Group A: patients with regular menstrual cycle (*n* = 22), including 4 patients in contraceptive therapy, 1 in contraceptive therapy and breastfeeding;Group B: patients with absence of menstrual cycle and less than 60 years old (*n* = 14), including 3 patients who underwent hysterectomy;Group C: patients with absence of menstrual cycle aged 60 years or more (*n* = 13), including 1 patient in hormonal replacement therapy.

### 2.2. Tissue Selection Process

Histologic diagnoses and immunohistochemistry were obtained at the Section of Anatomic Pathology, Department of Biomedical and Neuromotor Sciences, University of Bologna, at Bellaria Hospital, Bologna, Italy. All tissues were fixed in 4% buffered formalin and paraffin embedded according to routine protocol. Serial 2μm sections were obtained from each block and stained with Haematoxylin and Eosin (H&E) for histologic evaluation. 

Cases were retained for the present study when normal breast tissue was present around the lesion leading to surgery and the block containing the largest amount of normal breast tissue was selected for immunohistochemical studies. When possible, tissue obtained from the upper outer quadrant (UOQ) was chosen. Apocrine cysts, sclerosing adenosis and all the benign changes observed in aging breast were excluded from evaluation.

After histological evaluation on H&E, areas with at least 5 normal terminal ductular lobular units (TDLU) were selected for Tissue Micro-Arrays (TMA) construction. TMA were constructed following the technique described by Zimpfer et al. [19].

### 2.3. Tissue Immunohistochemical Evaluation

Immunohistochemical evaluation was made on TMA sections.

Evaluation and quantification of biomarkers was performed by counting the percentage of positive cells at 40x magnification. A minimum of 4 terminal-ductular-lobular units were evaluated for each marker. Immunohistochemical staining was performed on a Ventana Automatic Stainer (Ventana Medical Systems, Inc). The following pre-diluted antibodies were supplied by Ventana: Estrogen Receptor (ER) (clone SP1), Progesterone Receptor (clone 1E2), Androgen Receptor (clone SP107), Ki67 (clone 30-9) and Gross Cystic Disease Fluid Protein 15 (clone EP1582Y).

### 2.4. Statistical Analysis

For each patient who participated in the present study, the following data were collected: age, Body Mass Index (BMI), contraceptive therapy, post-menopausal hormonal replacement therapy, date of surgery, type of surgical procedure, and site and size of the lesion leading to surgery.

BMI was evaluated as a three-level categorical variable with cutoffs at 18.5 and 25 kg/m^2^.

The variability of expression of ER, PGR, AR, GCDFP-15 and Ki-67 markers was very limited in the myoepithelial and stromal cells; therefore, it was evaluated only in the epithelial cells. Due to the limited population size and the skewed distribution of most markers (Appendix A), and although the hypothesis of normal distribution was not always rejected by the Shapiro–Wilk test (Appendix A), median and interquartile range (IQR) were used as descriptive statistics. For each marker the percentage of patients with positive expression and the percentage of positive cells among positives were calculated. Comparisons of subgroups of patients according to the percentage of patients with positive expression were conducted using Fisher’s exact test when the expected frequency of each cell was < 5, or using chi-square otherwise. The differences in the percentage of positive cells across subgroups defined by menstrual cycle and nature of lesions were evaluated by Mann–Whitney U test, and those across phenotypes and BMI subgroups by Kruskal–Wallis test. Post hoc analyses by Dunn’s test with Holm adjustment for multiple comparisons were carried out after significant Kruskal–Wallis tests. Stata v.15.1 was used for all analyses, specifically the dunntest procedure [20] was used to perform the post hoc analyses. Statistical significance was set at *p* = 0.05.

## 3. Results

Descriptive statistics of the study population are reported in Table 1. The median age of patients was 50 years, with younger patients in group A and older patients in group C. Median BMI was lower in group A (22.81 kg/m^2^) and higher in group C (27.92 kg/m^2^). Most patients had a malignant diagnosis (75.0%), with a lower incidence in group A (61.9%). Only 5 patients, all in group A, were receiving contraceptive therapy and 1 patient (of group C) was under hormonal therapy.

The observed expression of epithelial markers is reported in Table 2. AR and GCDFP-15 were expressed in the large majority of patients (84.2% and 72.2%, respectively), while Ki-67 (38.6%) and Estrogen receptor alpha (43.2%) showed lower prevalence. 

GCDFP-15 and ER were the most evidenced markers (median rates of 55% and 36%, respectively). (Appendix A). 

Most of the breast cancers here were ER positive, with 16 cases being Luminal A, 13 cases Luminal B cancers (two of which HER2 enriched) and 4 cases triple negative (TNBC) [8,9].

Hormone expression variations according to age:

There were significant differences among groups, specifically the proportion of positive cases for PR and AR was lower in group C. As for the distribution of expression among positive patients (Table 3), women in group A showed significantly lower values of GCDFP-15 with respect to group C (median: 30 vs. 90) and borderline lower values of ER with respect to group B (median: 8.5 vs. 48.5).

Hormone expression variations according to the menstrual cycle:

The proportions of positives for each marker evaluated in the 22 patients of group A were not significantly different for menstrual cycle phase; however, the expression of PR and Ki67 was remarkably higher in women in follicular phase (41.5 vs. 18 and 8.25 vs. 2.5, respectively) and not far from reaching statistical significance (Table 4).

Hormone expression variations according to BMI:

ER positive cases increased with higher BMI, not significantly. PR positive cases were less frequent in the BMI ≥ 25 patients with respect to underweight and normal weight patients (33.3% vs. 75.0% and 73.7%; Fisher’s exact test: *p* = 0.028, Table 5); on the contrary, underweight women showed a higher, but not significant, proportion of Ki-67 positives. The median values of expression did not differ according to BMI.

Hormone expression according to the type of lesion leading to surgery (benign versus malignant):

The proportion of Ki-67 positives was higher among patients who showed benign lesions than malignant lesions (66.7% vs. 25.8%, *p* = 0.032). The amount of hormone receptor expression did not change significantly for any marker (Table 6). Specifically, ER and PR expression was similar in breast tissue adjacent to benign and malignant lesions. Most patients included in the present study were affected by ER positive breast cancers.

Examples of positivity obtained with ER, PR, AR and Ki67 are shown in Appendix A.

## 4. Discussion

The present study confirms the great variations in hormone receptor expression occurring in adult women. The data here shown reveal hormone receptors variations mainly according to age and BMI, while little changes were observed during the menstrual cycle. While ER, AR and GCDFP-15 were higher in post-menopausal patients, PR expression decreased as age increased, reaching very low values in patients older than 60 (Group C). 

The ER increase in older women shown here is consistent to the data published by Lawson et al. [21] who observed higher ER levels in post-menopausal compared to pre-menopausal women. 

The progressive increase in ER expression in older women lead to some considerations about the ER role in breast cancer development. Breast cancer is known to have a peak of incidence in the 6th decade of life. Moreover, hormonal receptor positive breast cancers, classified as Luminal A or Luminal B, according to the St. Gallen definition [8,9], are the most frequent cancer types encountered in elderly women [22]. 

The association between hormonal expression and cancer has been studied extensively. Khan et al. [14] observed a significant ER expression increase in normal breast tissue of patients who underwent surgery for breast cancer. Steroid hormone receptors play an important role in regulating cell cycle and cell proliferation [23]. In Luminal A and B breast cancer, ER binds to the CCND1 promoter favoring cell proliferative activity. The ER higher expression here observed in post-menopausal women can lead to ER-driven transcription of CCND1, that is a crucial factor in neoplastic transformation [23].

ER expression in our study showed a peak in group C (patients older than 60 y/o) while PR gradually reduces after the 6th decade of life, supporting the concept that estrogen plays a role in developing Luminal A and Luminal B cancers (typically ER+ and PR +/−). Moreover, in the present study, ER mean values observed in normal breast epithelial cells surrounding cancers were slightly higher than those found in normal breast tissue surrounding benign breast lesions.

Similar considerations can be done for AR. AR is expressed in the majority of breast cancers [24,25,26]. Cancers being ER/PR/HER2 negative but AR positive showed better outcome compared to those AR negative [24,25]. AR expression in our study was higher in older patients and significantly lower in younger patients. AR positive breast cancers, are also of apocrine histotype [27]. This finding is consistent with the present data that AR expression is paralleled by GCDFP-15 expression. GCDFP-15 is strongly expressed in Apocrine carcinomas which are well known to be AR positive [27,28]. This suggests that AR could promote the development of Apocrine carcinomas in post-menopausal women [24,25].

Ki67 showed higher expression in younger patients. Ki67, an antigen expressed in cells G1, S, G2 and M phases, is widely used in daily practice as proliferation marker. Higher levels in Ki67 in normal breast tissue from younger patients may be justified by a higher regenerating tissue levels under the influence of the periodic hormonal variation of the menstrual cycle. 

Several studies demonstrated that a BMI greater than 25 kg/m^2^ represents a risk factor for the development of breast cancer [29]. The risk of breast cancer raises significantly in obese women (BMI > 35 kg/m^2^) compared to those having a BMI within normal ranges [6]. Estrogen circulating levels are significantly higher in overweight and obese patients which usually develop ER+ cancers; this, together with the evidence that expression of ER and PR levels is significantly higher in obese patients’ breast cancers, led to the conclusion that estrogens could play a role in breast cancerogenesis [6,29,30]. In the present series, ER expression showed an increasing trend according to BMI even if it did not reach statistical significance. It should be underlined that, in our series women with BMI < 18.5 were predominantly in the group A (mean age < 42 y/o) while women in overweight group belonged from groups B (mean age 54 y/o) and C (mean age 66 y/o). Therefore, the increased ER expression could be related to the older age and not only to increasing BMI. 

The limited number of young obese or overweight patients, in the pre-menopausal period, does not allow definitive conclusions to be drawn.

In the present study, hormonal expression variation according to menstrual cycle phase was not so evident as expected. The present data demonstrated a tendency for reduced hormonal expression from follicular to luteal phase. Data here shown, even if not reaching statistical significance, are in keeping with those of Battersby et al. [31] who observed a marked reduction in ER expression during the menstrual cycle, while PR did not show a significant variation. The same result was achieved by Khan et al. [32]. 

Among the limitations of the current study, its limited sample size prevented us from obtaining results with robust statistical significance; therefore, our findings should be carefully interpreted.

## 5. Conclusions

Our study highlighted a high variability in the expression of hormonal receptors in healthy breast epithelial cells. The combination of different expressions of ER, PR, AR, GCDP-15 and Ki67 could be a risk factor in the development of breast cancer. Even the molecular subtypes of breast cancer could be influenced by the normal expression of hormones at a certain age: for example, triple negative cancer are more frequent at younger age when we demonstrated that ER expression is at its lowest value. On the contrary, post-menopausal women’s breasts, characterized by higher expression of ER, PR and AR in the epithelial component, tend to develop ER+/PR+/AR+ cancers.

## Figures and Tables

**Table 1 jpm-11-00387-t001:** Characteristics of the study population.

	Study Population	Group A	Group B	Group C
Age; median (IQR)	50 (17)	42 (8)	54 (7)	66 (10)
BMI; median (IQR)	23.92 (4.7)	22.81(2.9)	25.39 (3.9)	27.92 (5.5)
Malignant; n (%)	36 (75.0)	13(61.9)	12 (85.7)	11 (84.6)

**Table 2 jpm-11-00387-t002:** Epithelial expression of markers.

	Positive (%)	Median	Interquartile Range
ER	43.2	36	41.5
PGR	56.1	23	37
AR	84.2	30	26.5
GCDFP-15	72.2	55	70
Ki-67	38.6	3	6

**Table 3 jpm-11-00387-t003:** Expression of hormone receptors according to age.

	Group A (%; Median)	Group B (%; Median)	Group C (%; Median)	*p*-Value	Post hoc Comparisons
ER	31.8%; 8.5	54.5%; 48.5	54.5%; 43	2.32; 0.314 *	5.9; 0.050; (none)
PR	85.0%; 27	55.6%; 16	8.3%; 2	<0.001	2.08; 0.354
AR	94.4%; 30	100.0%; 33.5	50.0%; 30	0.005	0.42; 0.809
GCDFP-15	57.9%; 30	85.7%; 65	90.0%; 90	0.157	9.42; 0.009 *;(A < C)
Ki-67	45.0%; 8	38.5%; 2	27.3%; 3	0.673	2.06; 0.357

* χ^2^-test.

**Table 4 jpm-11-00387-t004:** Expression of hormone receptors according to menstrual cycle.

	Follicular Phase	Luteinic Phase	*p*-Value	Test; *p*-Value
ER	40.0%; 16.5	33.3%; 8.5	0.570	0.71; 0.475
PR	88.9%; 41.5	88.9%; 18	1.000	1.68; 0.093
AR	100.0%; 34	85.7%; 35	0.467	−0.39; 0.698
GCDFP-15	75.0%; 20	44.4%; 35	0.335	−0.86; 0.389
Ki-67	25.0%; 8.25	44.4%; 2.5	0.620	1.88; 0.060

**Table 5 jpm-11-00387-t005:** Expression of hormone receptors according to BMI.

	BMI < 18.5	18.5 ≤ BMI < 25	BMI ≥ 25	*p*-Value	Test; *p*-Value
ER	20.0%; 3	40.9%; 36	52.9%; 48	0.423	1.64; 0.440
PR	75.0%; 16	73.7%; 32	33.3%; 19	0.028	0.43; 0.808
AR	100.0%; 16	85.7%; 34	75.0%; 30	0.568	1.51; 0.469
GCDFP-15	50.0%; 70	77.8%; 30	71.4%; 85	0.516	3.09; 0.214
Ki-67	80.0%; 5	38.1%; 3	27.8%; 3	0.104	0.74; 0.692

**Table 6 jpm-11-00387-t006:** Expression of hormone receptors according to the nature of the treated lesion.

	Benign	Malignant	*p*-Value	Test; *p*-Value
ER	41.7%; 48	45.2%; 32	0.04; 0.836 *	1.11; 0.266
PR	63.6%; 16	51.7%; 27	0.723	0.21; 0.832
AR	90.9%; 22.5	80.8%; 30	0.646	−1.57; 0.117
GCDFP-15	54.6%; 25	79.2%; 70	2.24; 0.134 *	−1.28; 0.202
Ki-67	66.7%; 4	25.8%; 3	0.032	0.48; 0.630

* χ^2^-test.

## Data Availability

The data presented in this study are available on request from the corresponding author.

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
