# Peer review of "Hormone Receptor Expression Variations in Normal Breast Tissue: Preliminary Results of a Prospective Observational Study"

_jpm, 2021, doi:10.3390/jpm11050387_

Round 1

Reviewer 1 Report

Santadrea et al. focused on hormone receptor variations in breast normal tissue. The authors would assess ER, PR, AR, GCDFP-15 and Ki-67 markers in adult breast normal tissue. Although the relevance of this manuscript could be high, this observational study has a sample size too low to achieve a statistically robust design and for generalization of findings. Thereby, in my opinion, this manuscript needs of substantial revisions. My suggestion is to specify also in the title the observational/exploratory nature of the study. Introduction lacks of recent literature findings, the aim of the work it is unclear.

Statistical analyses were carried on a cohort of 49 patients stratified into three groups in accordance to primary variables such as menstrual cycle and age. Methods lack of experimental details necessary to reproduce the analyses. Normality inspection test and distribution of variables should be included as Supplementary to justify non-parametric descriptive statistics. Discussion is generic and poor, lacking of recent literature findings.

Author Response

  • Santadrea et al. focused on hormone receptor variations in breast normal tissue. The authors would assess ER, PR, AR, GCDFP-15 and Ki-67 markers in adult breast normal tissue. Although the relevance of this manuscript could be high, this observational study has a sample size too low to achieve a statistically robust design and for generalization of findings. Thereby, in my opinion, this manuscript needs of substantial revisions. My suggestion is to specify also in the title the observational/exploratory nature of the study.

Thank you for your suggestion.

Our preliminary results are the early observations that emerged from an ongoing prospective analysis that started in 2015.

Due to COVID-19 pandemic, we were forced to stop our essays on normal breast tissue, but we will resume them as soon as possible. We understand and agree that we are presenting a small sample of patients, and that it needs to reach a higher statistic impact to be an incisive contribution. Nevertheless, we wanted to contribute to this Special Issue with our preliminary observations, waiting to enlarge our cohort.

We proceeded to specify in the title and in the text that this is a preliminary observational analysis.

  • Introduction lacks of recent literature findings, the aim of the work it is unclear.

Thank you. We reviewed the introduction of our paper, adding recently published data and presenting more clearly the current evidence regarding the expression of hormonal receptors in breast tissue and the aim of our study.

  • Statistical analyses were carried on a cohort of 49 patients stratified into three groups in accordance to primary variables such as menstrual cycle and age. Methods lack of experimental details necessary to reproduce the analyses.

Thank you for your suggestion. We reviewed statistical analysis in methods and added the requested supplementary material  

    •  
    • Normality inspection test and distribution of variables should be included as Supplementary to justify non-parametric descriptive statistics. Discussion is generic and poor, lacking of recent literature findings.

Unfortunately most of the work published on normal breast is rather old, and it is difficult to find recent papers. When possible old references were eliminated and substituted with more recent ones. (previous references 5 and 6, substituted with the following: 5)       Daly AA, Rolph R, Cutress RI, Copson ER. A Review of Modifiable Risk Factors in Young Women for the Prevention of Breast Cancer. Breast Cancer (Dove Med Press). 2021 Apr 13;13:241-257. doi: 10.2147/BCTT.S268401. PMID: 33883932; PMCID: PMC8053601)

In addition, we tried to improve the discussion sections, according also to Reviewer’s 2 suggestions.

Reviewer 2 Report

The topic is interesting and the paper is well written but statistical design is not appropriate. Total number of patients is very low, in particular for group B and C. The study could be interesting, but with a larger number of patients.

Author Response

  • The topic is interesting and the paper is well written but statistical design is not appropriate. Total number of patients is very low, in particular for group B and C. The study could be interesting, but with a larger number of patients.

Thank you for your suggestion.

Our preliminary results are the early observations that emerged from an ongoing prospective analysis. Due to COVID-19 pandemic, we were forced to stop our essays on normal breast tissue, but we will resume them as soon as possible. We understand and agree that we are presenting a small sample of patients, and that it needs to reach a higher statistic impact to be an incisive contribution. Nevertheless, we wanted to contribute to this Special Issue with our preliminary observations, waiting to enlarge our cohort. We proceeded to specify in the title and in the text that this is a preliminary observational analysis.

Reviewer 3 Report

Single sentences do not need to be paragraphs, combine and present the introduction as a coherent entity instead of segmented.

Result tables are cluttered with a lot of values that could be made into new tables or organized in a better fashion. Also some font discrepancies in Table 2.

Have consistency with capitalizing "Table" when referring to a results table

Should present rough baseline (healthy) receptor number values to be able to more objectively compare increases or decreases in receptor numbers

Did this lab complete the data shown in Figure S1? The introduction says that one of the aims of the study is to use IHC to address the receptor values in different populations, yet IHC is not mentioned after the methods and Figure S1 is not mentioned or analyzed in results or discussions. As the aim of the study, this has to be incorporated.

In discussion, further assess specific relationship between ER level increases in the elderly compared with the incidence of the most common (Luminal A and B) cancer types.

there are references in the discussion that should be in the introduction based on the fact that they are presented and then not supported by data that was discovered by the researchers (sick lobe theory and young ER receptors and triple negative cancer)

"increase in weight >5%" compared to what? over what period of time does this weight increase then indicate an increased chance of breast cancer.

indicate that one of the downfalls of the study was not having a young, overweight population to examine for ER expression

conclusion doesn't really address what the paper accomplished, it more so discusses facts that were understood before reading the paper.

I appreciate the idea of the paper but it needs reworking. Data is there but presented in an unorganized fashion that could be improved by separating tables a bit or maybe adding a sporadic graph or two. The beginning addresses the importance of IHC to the study and then then paper does not further discuss IHC. The introduction is adequate but there could definitely be more information presented, such as some of the material in the discussion. The discussion is lacking an in depth analysis or explanation of the specific results, rather it is a fairly general passover of some of the data that is then briefly explained. As a whole, I do not believe that other studies are required per se, but that the piece needs to have a deeper analysis into the data that is present. In addition, a lot of english/grammatical review combined with organizational efforts.

I do very much like the idea behind this paper, looking to address breast cancer in a way that could lead to more preventative measures instead of trying to determine treatments post-diagnosis. The study could've used more participants but that is an understandably difficult metric to achieve. I commend that authors work thus far and look forward to seeing what they have to offer going forward.

Author Response

  • Single sentences do not need to be paragraphs, combine and present the introduction as a coherent entity instead of segmented.

Thank you. We reviewed the whole introduction section.

  • Result tables are cluttered with a lot of values that could be made into new tables or organized in a better fashion.

Thank you. We reviewed and “lightened” Tables in order to make them easier to understand.

  • Also some font discrepancies in Table 2.

Proceeded to review Table 2 font

  • Have consistency with capitalizing "Table" when referring to a results table

Capitalized the word “Table” in the text, when referred to a result table

  • Should present rough baseline (healthy) receptor number values to be able to more objectively compare increases or decreases in receptor numbers

As aim of the study is to evaluate hormone receptors variations in normal tissue, all the present data are obtained in normal glands.

  • Did this lab complete the data shown in Figure S1? The introduction says that one of the aims of the study is to use IHC to address the receptor values in different populations, yet IHC is not mentioned after the methods and Figure S1 is not mentioned or analyzed in results or discussions. As the aim of the study, this has to be incorporated.

Yes, the immunohistochemical staining were performed in our lab, as explained in the Materials and Methods section.

The following sentence: “Examples of positivity obtained with ER, PR, AR and Ki67 are shown in figure S1.” Has been added at the end of the Results section.

  • In discussion, further assess specific relationship between ER level increases in the elderly compared with the incidence of the most common (Luminal A and B) cancer types.

The following sentence together with a reference have been added:

“Steroid hormone receptors play an important role in regulating cell cycle and cell proliferation [19]. In Luminal A and B breast cancer, ER binds to the CCND1 promoter favoring cell proliferative activity. The ER higher expression here observed in post-menopausal women can lead to ER-driven transcription of CCND1, that is a crucial factor in neoplastic transformation [19].”

  • There are references in the discussion that should be in the introduction based on the fact that they are presented and then not supported by data that was discovered by the researchers (sick lobe theory and young ER receptors and triple negative cancer)
    • The Introduction section has been modified and the references moved accordingly.
  • "increase in weight >5%" compared to what? over what period of time does this weight increase then indicate an increased chance of breast cancer. indicate that one of the downfalls of the study was not having a young, overweight population to examine for ER expression conclusion doesn't really address what the paper accomplished, it more so discusses facts that were understood before reading the paper.

These sentences refer to the data reported in references quoted. Nevertehelss for the sake of clarity, the sentence reporting “increase weight >5%” was deleted. In addition, the following sentence has been added: “The limited number of young obese or overweight patients, in the pre-menopausal period, does not allow definitive conclusions to be drawn.”

Round 2

Reviewer 1 Report

The authors attempt to clarify better my concerns. 

In my opinion, the current version has been largely improved. 

Reviewer 2 Report

The low statistical impact due to limited patient's number is well clarified in the text. This point is the main limitation of this work. Nevertheless I think that this preliminary results are well presented and interesting for the readears of this Specia Issue. 

Reviewer 3 Report

good changes regarding tables and results. overall acceptable despite small patient population which is understandable due to COVID19